# Association between waist circumference and sleep disorder in the elderly: Based on the NHANES 2005–2018

**Yuting Zhong[1], Ying Li[1], Maolin Zhong[1], Cheng Peng[2], Hui Zhang[3], Kejun Tian●[4]***

**1** Department of Anesthesiology, First Affiliated Hospital of Gannan Medical University, Ganzhou, Jiangxi, China, **2** Department of Anesthesiology, Ganzhou People's Hospital, Ganzhou, Jiangxi, China, **3** Department of Experimental Medicine, University of Rome Tor Vergata, Rome, Italy, **4** Department of Cardiology, First Affiliated Hospital of Gannan Medical University, Ganzhou, Jiangxi, China

* tiankejun371@163.com

**Data Availability Statement:** The datasets given in this investigation are accessible through the online repositories (https://www.cdc.gov/nchs/nhanes/index.htm).

## Abstract

The existing data do not consistently support the link between elderly adults' waist circumferences and sleep disorders. This study aimed to evaluate whether waist circumference was connected with sleep disorder in the elderly. This cross-sectional study utilized data from the 2005–2018 National Health and Nutrition Examination Survey (NHANES) regarding waist circumference, sleep disorders, and confounding factors. Included in the study were participants older than 60 who completed sleep questionnaires and waist circumference measurements. Using a multivariate logistic regression model and subgroup analyses, the relationship between waist circumference and sleep disorder was evaluated. To explore the non-linear relationship, restricted cubic spline (RCS) with three knots coupled with a logistic regression model to assess the dose-response relationship between waist circumference (continuous variables) and sleep disorder. A total of 2,545 (Weighted 14,682,916.3) elderly participants with complete information were included in the analysis and 312 (Weighted 1,777,137.8) subjects met the definition of sleep disorder. Compared with participants without sleep disorder, those with sleep disorder had a higher waist circumference (100.80 cm *vs.* 108.96 cm, *P* < 0.001). The results of the multivariable adjusted logistic regression model suggested that those in quartiles 4 ($\geq$ 75th percentile) for their waist circumference had higher odds of sleep disorder [adjusted odds ratio (AOR) = 2.75, 95% confidence interval (CI) = 1.66–4.54, *P* < 0.001] compared with those in quartile 1. The RCS result showed that the OR of sleep disorder and waist circumference displayed a linear relationship (*P* < 0.001, Non-linear *P* = 0.642). Age and gender subgroup analysis revealed comparable relationships between waist circumference and sleep disorder among elderly individuals. Waist circumference was associated with sleep disorders in the elderly. There was a dose-response relationship between waist circumference and the likelihood of sleep disorder. Those with a larger waist circumference were more likely to have a sleep disorder than those with a smaller waist circumference.

**Funding:** The author(s) received no specific funding for this work.

**Competing interests:** The authors have declared that no competing interests exist.

## Introduction

Sleep disorders are prevalent among the elderly, and they can have a substantial impact on their quality of life and overall health [1, 2]. Approximately fifty percent of the elderly, according to statistics, frequently experience sleep problems in their daily lives [3]. Previous studies have shown that sleep disorders are associated with a range of adverse outcomes, including cognitive impairment [4], depression [5, 6], cardiovascular disease [7], and lower urinary tract symptoms [8]. In addition, the proportion of elderly people is increasing due to increased life expectancy and enhanced socioeconomic development [9]. Examining prospective risk factors for sleep disorders may therefore facilitate sleep disorder prevention and intervention, thereby enhancing the quality of life of the elderly and decreasing the disease burden on society.

Recent interest in the potential function of body composition measures in the development of sleep disorders has increased [10]. A substantial amount of research has focused on exploring the relationship between obesity and sleep disorder, highlighting the bidirectional nature of this association [11, 12]. Although body mass index (BMI) has been used as an important indicator for assessing obesity, waist circumference is a more sensitive predictor of health risks associated with obesity than BMI [13]. Waist circumference is a common indicator of central adiposity and has been associated with a variety of adverse health outcomes, such as cardiovascular disease, metabolic syndrome, and type 2 diabetes [14]. However, the association between waist circumference and sleep disorder in the elderly population remains ambiguous.

Given the high prevalence of sleep disorders in the geriatric population and the potential influence of waist circumference on sleep, additional research is required in this area. National Health and Nutrition Examination Survey (NHANES) is a complex, multistage probability design sample of the noninstitutionalized U.S. population. Using NHANES data, the purpose of this investigation was to evaluate the association between waist circumference and sleep disorder in the elderly. This study's findings could have significant implications for the prevention and treatment of sleep disorders in the geriatric population.

## Methods

### Study population

The National Health and Nutrition Examination Survey (NHANES) is a nationally representative survey of the U.S. population that provides extensive information about the nutrition and health of the general U.S. population. The NHANES uses a complex, multistage probability sampling design to ensure that the data collected is representative of the civilian, non-institutionalized population of the U.S. [15]. The NHANES survey data are publicly accessible to data researchers and consumers. The National Center for Health Statistics (NCHS) collects its data in biennial cycles [16]. To acquire large samples for analysis, seven cycles of continuous NHANES data from 2005 to 2018 were combined. The protocol for the survey was endorsed by the NCHS Research Ethics Review Board, and all participants provided written informed consent. There is more information about the NHANES available at http://www.cdc.gov/nhanes.

Of 70,190 participants extracted from the NHANES database, we excluded those with age < 60 years (n = 56,710), missing information on sleep disorder (n = 4,071), and other covariates (n = 6,864). Finally, a total of 2,545 participants were included in this study. The flow chart of the systematic selection process is shown in Fig 1.

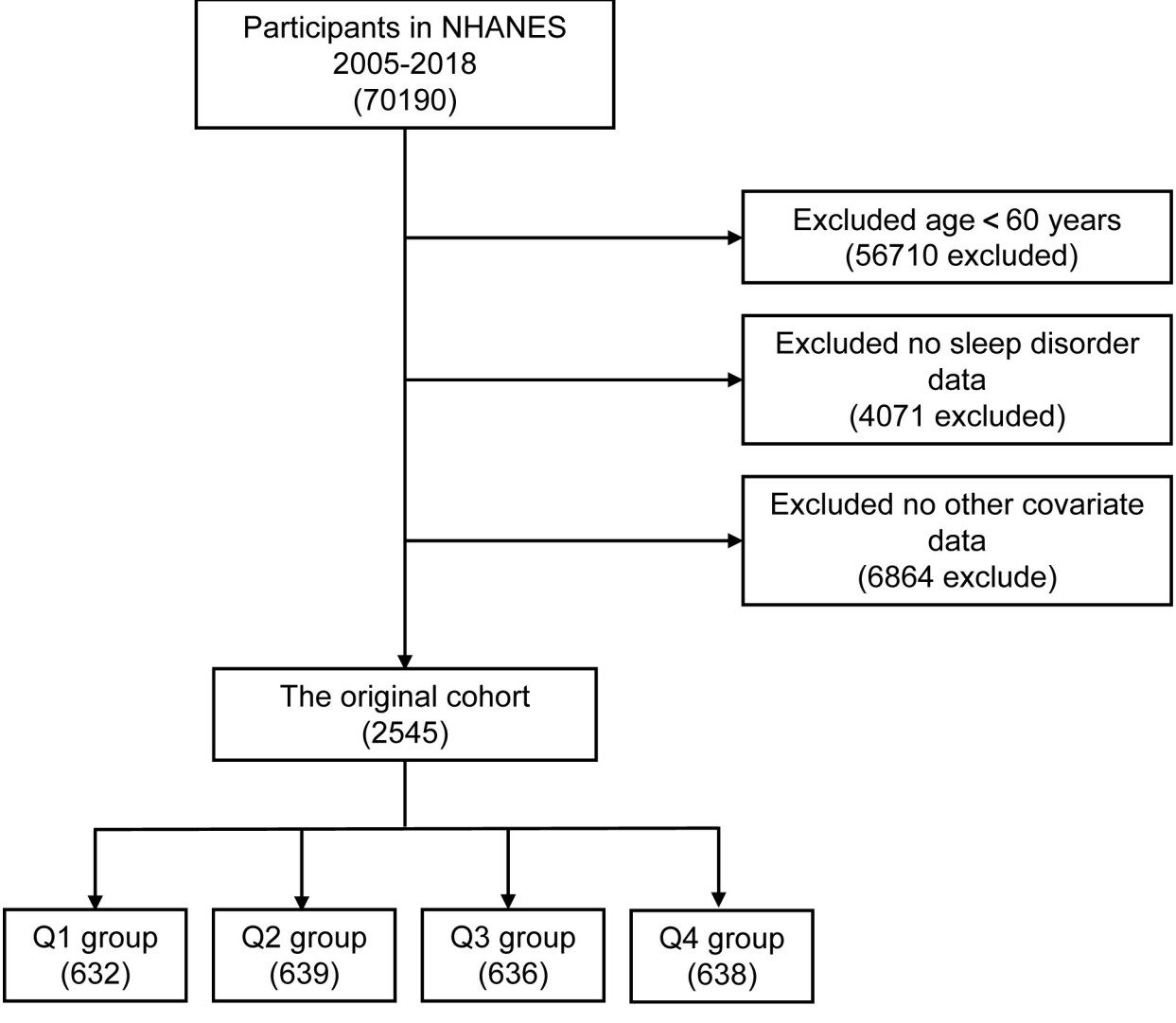

**Fig 1. Screening of admissions for inclusion.**

### Main independent variable

All waist circumference is taken at the mobile examination center (MEC). We divided the waist circumference of the subjects into groups of Q1 ($< 92.5$cm), Q2 (92.5–101.3cm), Q3 (101.3–110.8cm) and Q4 ($\geq 110.8$cm) according to quartile.

### Covariates

We incorporated the following covariates derived from NHANES interview, examination, laboratory, and questionnaire data: Age, gender, race, education level, marital status, family income (poverty income ratio, PIR), alcohol intake, smoking status, and recreational physical activity. Age was categorized into three groups: 60–69 years, 70–79 years, and 80+ years. Race was categorized into five groups: Mexican American, non-Hispanic Black, non-Hispanic White, other Hispanic, and other/multiracial. Education level was assessed by the question "What is the highest grade or level of school you have completed or the highest degree you have received?" [less than high school/high school graduate or above]. Participants were divided into two categories for marital

status (have spouse/without spouse). Based on the original survey records, PIR was assessed. Participants were divided into four categories for alcohol intake (non-drinker, 1 to < 5 drinks/month, 5 to < 10 drinks/month, or 10+ drinks/month). Participants were asked whether they had ever smoked 100 cigarettes in their lifetime and whether they smoked currently to identify current and former smokers. Participants were defined as former smokers if they did not smoke currently but had ever smoked 100 cigarettes in the past. We also included total daily calories as a potential covariable. Recreational physical activity was assessed using the "Physical Activity" questionnaire. Within this questionnaire, participants were queried about the frequency of engaging in vigorous work activity, the number of days they participated in vigorous or moderate recreational activities, and the duration of vigorous-intensity work and moderate-intensity recreational activities in minutes. According to the Physical Activity Guidelines, it is recommended that individuals engage in at least 75 minutes of vigorous exercise per week or 150 minutes of moderate exercise per week. Doctors diagnosed diseases, including diabetes and stroke, basking participants "Have you ever been told by a doctor or health professional that you have __?". Comorbidity included 1) congestive heart failure, 2) chronic obstructive pulmonary disease (emphysema and/or chronic bronchitis), 3) coronary artery disease, 4) cancer and 5) hypertension [17]. The comorbidity index indicates how many complications an individual has.

## Outcome variable

The outcome variable was that the participant had sleep disorder. Data were collected in participants' homes by interviewers using a computer-assisted personal interview (CAPI) system. Various questions concerning sleep patterns were asked, including usual sleep time on weekdays or workdays, usual wake time on weekdays or workdays, sleep hours, how often participants snore, snort, or stop breathing, if participants have ever told a doctor, they had trouble sleeping, and how often they feel overly sleepy during the day. In our study, the NHANES variable SLQ050 (ever told a doctor or other health professional that you had trouble sleeping) was used to determine if a participant had a sleep disorder [18].

## Statistical analysis

The intricate sampling methodology employed, as detailed in reference [19], encompasses stratified, cluster, and multistage sampling, along with unequal probability sampling proportional to a measure of size (PPS). This approach necessitates the consideration of sampling weights. This design allows for the integration of additional cycles, thereby enhancing statistical reliability (WTDRD1/7). However, conventional regression methods are inadequate as they can yield incorrect inferential outcomes. Specifically, the standard error and confidence intervals of parameter estimates may be significantly underestimated, and the probability of Type I errors in hypothesis testing is substantially elevated. Consequently, we used SURVEYMEANS, SURVEYREG, and SURVEYLOGISTIC to perform accurate statistical descriptions and logistic regression analyses that account for complex sampling designs.

Continuous variables are presented as weighted medians with interquartile ranges (IQRs), while categorical variables are displayed as weighted numbers and weighted percentages. Group comparisons were performed using the $\chi^2$ test for categorical variables and the Mann-Whitney U test for continuous variables, as appropriate. Waist circumference was categorized into four groups. Weighted multivariate logistic regression analyses were conducted to explore the relationship between waist circumference and sleep disorders. In the unadjusted Model 1, no confounders were controlled. Model 2 adjusted for gender and age. Beyond the adjustments made in Model 2, additional potential confounding factors were accounted for in further analyses. To explore the non-linear relationship, weighted restricted cubic spline (RCS)

with three knots coupled with a logistic regression model to assess the dose-response relationship between waist circumference (continuous variables) and sleep disorder. Subgroup analyses stratified by age, gender and race were performed. All statistical analyses were performed using R software (version 4.1.1), and $P < 0.05$ was considered statistically significant.

## Results

### Description of the study population

A total of 2,545 (Weighted 14,682,916.3) participants aged 60–85 years who had measured data for waist circumference and sleep disorder were included in this study. Of the included participants, 52.6% (weighted 7,725,328.4) were 60–69 years old, followed by 70–79 years old (27.1%, weighted 3,982,584.6), and 80+ years old (20.3%, weighted 2,975,003.3). The gender distribution was relatively equal. Participants with sleep problems made up more than a tenth of the study population (12.1%). Gender, waist circumference, smoking status, recreational physical activity, diabetes, stroke, and comorbidity index were significantly different in the sleep disorder group compared with the non-sleep disorder group ($P$ <0.05, Table 1). Participants with sleep disorder were more likely to be male, larger waist circumference, higher percentage of current smokers, lower percentage of recreational physical activity, had a history of diabetes and stroke, and higher comorbidity index.

### Baseline comparison of different waist circumference groups

Of the participants included in the study, 632 participants had a waist circumference < 92.5 cm (Q1 group), 639 participants were between 92.5 and 101.3 cm (Q2 group), 636 participants were between 92.5 and 110.8 cm (Q3 group), and 638 patients were ≥ 110.8 cm (Q4 group). Table 2 lists the weighted demographic features of all participants between different waist circumference groups. In different groups of waist circumference, age, gender, race, marital status, alcohol intake, recreational physical activity, diabetes, and comorbidity index are significantly different ($P$ <0.05, Table 2).

### Association of waist circumference with sleep disorder

The incidence of sleep disorder was significantly higher among patients with Q4 group (21.2%; weighted 815,257.5), compared with patients with Q3 group (12.0%; weighted 452,613.4), Q2 group (8.3%; weighted 288,324.0), and Q1 group (6.2%; weighted 220,942.9; $P$ <0.001, Table 2). We have used three weighted multivariate logistic regression models to show the relationship between waist circumference with sleep disorder in Table 3: model 1, no covariate was adjusted; model 2, Gender, waist circumference, smoking status, recreational physical activity, diabetes, stroke, and comorbidity index were adjusted. We found a significantly positive association between waist circumference with the incidence of sleep disorder in the unadjusted model (Model 1). In model 2, the ORs (95% CIs) after adjusting for age and gender for incidence of sleep disorder in participants with Q4 group compared with those with Q1 group was 3.79 (2.28–6.33) (Table 3, $P < 0.001$). In model 3, the ORs (95% CIs) after adjusting for related indexes for incidence of sleep disorder in participants with Q3 group compared with those with Q1 group was 2.75 (1.66–4.54) (Table 3, $P < 0.001$).

### The dose-response relationship of waist circumference and the risk of sleep disorder

A dose-response relationship was examined between waist circumference and the risk of sleep disorder. The result showed that the OR of sleep disorder and waist circumference displayed a linear relationship ($P$ <0.001, Non-linear $P = 0.642$), as shown in Fig 2.

**Table 1. Baseline characteristics of subjects in the different groups[†].**

| Covariates | No sleep disorder group | Sleep disorder group | P |
|---|---|---|---|
| **Unweighted number** | 2233 | 312 | |
| **Weighted number** | 12905778.5 | 1777137.8 | |
| **Age** | | | 0.145 |
| 60–69 years | 6685232.6 (51.8%) | 1040095.8 (58.5%) | |
| 70–79 years | 3514605.7 (27.2%) | 467978.9 (26.3%) | |
| 80+ years | 2705940.2 (21.0%) | 269063.1 (15.1%) | |
| **Gender** | | | 0.009 |
| Female | 7069189.2 (54.8%) | 787358.0 (44.3%) | |
| Male | 5836589.3 (45.2%) | 989779.7 (55.7%) | |
| **Race** | | | 0.950 |
| Mexican American | 465484.6 (3.6%) | 61731.6 (3.5%) | |
| Non-Hispanic Black | 1089543.8 (8.4%) | 139634.4 (7.9%) | |
| Non-Hispanic White | 10137245.4 (78.5%) | 1412115.5 (79.5%) | |
| Other Hispanic | 500958.3 (3.9%) | 60609.0 (3.4%) | |
| Other/multiracial | 712546.4 (5.5%) | 103047.3 (5.8%) | |
| **Education** | | | 0.883 |
| Less than high school | 4985773.9 (38.6%) | 676993.7 (38.1%) | |
| More than high school | 7920004.7 (61.4%) | 1100144.1 (61.9%) | |
| **Waist circumference (cm%)** | 100.80 (92.00, 110.00) | 108.96 (100.14, 121.30) | < 0.001 |
| **Marital Status** | | | 0.300 |
| Have spouse | 8331435.0 (64.6%) | 1217568.2 (68.5%) | |
| Without spouse | 4574343.6 (35.4%) | 559569.5 (31.5%) | |
| **PIR** | 3.09 (1.63, 5.00) | 2.97 (1.61, 4.94) | 0.584 |
| **Alcohol intake** | | | 0.267 |
| Non-drinker | 3595277.3 (27.9%) | 454446.7 (25.6%) | |
| 1–5 drinks/month | 6053762.6 (46.9%) | 983093.1 (55.3%) | |
| 5–10 drinks/month | 2581095.4 (20.0%) | 271132.7 (15.3%) | |
| 10+ drinks/month | 675643.3 (5.2%) | 68465.3 (3.9%) | |
| **Smoking status** | | | 0.036 |
| Never | 11688709.0 (90.6%) | 1517530.1 (85.4%) | |
| Current smoker | 1174006.2 (9.1%) | 258725.3 (14.6%) | |
| Former smoker | 43063.4 (0.3%) | 882.3 (0.0%) | |
| **Total calories (cal)** | 1798.47 (1411.43, 2261.06) | 1791.00 (1396.98, 2223.51) | 0.628 |
| **Recreational physical activity** | 6107095.6 (47.3%) | 683207.9 (38.4%) | 0.038 |
| **Diabetes** | 2234468.2 (17.3%) | 609371.4 (34.3%) | < 0.001 |
| **Stroke** | 738868.6 (5.7%) | 233567.4 (13.1%) | 0.001 |
| **Comorbidity index** | | | < 0.001 |
| 1 | 5444623.6 (42.2%) | 589068.4 (33.1%) | |
| 2 | 3935332.8 (30.5%) | 303065.5 (17.1%) | |
| 3 or greater | 3525822.2 (27.3%) | 885003.8 (49.8%) | |

[†]Percentage and median (Q25, Q75) were weighted.

*P<0.05

**Table 2. Baseline characteristics of subjects in the different waist circumstance groups.**

| Covariates | Q1 | Q2 | Q3 | Q4 | P |
|---|---|---|---|---|---|
| **Unweighted number** | 632 | 639 | 636 | 638 | |
| **Weighted number** | 3591579.4 | 3494513.6 | 3759868.2 | 3836955.1 | |
| **Age** | | | | | < 0.001 |
| 60–69 years | 1907208.5 (53.1%) | 1718810.8 (49.2%) | 1889078.5 (50.2%) | 2210230.6 (57.6%) | |
| 70–79 years | 756646.3 (21.1%) | 1019364.4 (29.2%) | 1109470.7 (29.5%) | 1097103.3 (28.6%) | |
| 80+ years | 927724.6 (25.8%) | 756338.4 (21.6%) | 761319.0 (20.2%) | 529621.3 (13.8%) | |
| **Gender** | | | | | < 0.001 |
| Female | 2554924.3 (71.1%) | 2004008.3 (57.3%) | 1724916.4 (45.9%) | 1572698.2 (41.0%) | |
| Male | 1036655.1 (28.9%) | 1490505.3 (42.7%) | 2034951.8 (54.1%) | 2264256.9 (59.0%) | |
| **Race** | | | | | < 0.001 |
| Mexican American | 89999.3 (2.5%) | 154649.3 (4.4%) | 123907.9 (3.3%) | 158659.7 (4.1%) | |
| Non-Hispanic Black | 297184.7 (8.3%) | 282430.3 (8.1%) | 279783.1 (7.4%) | 369780.1 (9.6%) | |
| Non-Hispanic White | 2692869.8 (75.0%) | 2678024.3 (76.6%) | 3104775.9 (82.6%) | 3073690.9 (80.1%) | |
| Other Hispanic | 133552.5 (3.7%) | 173892.2 (5.0%) | 159441.9 (4.2%) | 94680.7 (2.5%) | |
| Other/multiracial | 377973.0 (10.5%) | 205517.5 (5.9%) | 91959.6 (2.4%) | 140143.7 (3.7%) | |
| **Education** | | | | | 0.853 |
| Less than high school | 1322423.2 (36.8%) | 1346357.7 (38.5%) | 1478791.7 (39.3%) | 1515194.9 (39.5%) | |
| More than high school | 2269156.1 (63.2%) | 2148155.9 (61.5%) | 2281076.5 (60.7%) | 2321760.2 (60.5%) | |
| **Marital Status** | | | | | 0.046 |
| Have spouse | 2174768.6 (60.6%) | 2360078.2 (67.5%) | 2623667.1 (69.8%) | 2390489.2 (62.3%) | |
| Without spouse | 1416810.8 (39.4%) | 1134435.4 (32.5%) | 1136201.1 (30.2%) | 1446465.9 (37.7%) | |
| **PIR** | 3.22 (1.70, 5.00) | 3.04 (1.70, 5.00) | 3.30 (1.58, 5.00) | 2.77 (1.53, 4.76) | 0.412 |
| **Alcohol intake** | | | | | < 0.001 |
| Non-drinker | 1100069.4 (30.6%) | 1048987.3 (30.0%) | 987660.3 (26.3%) | 913006.9 (23.8%) | |
| 1–5 drinks/month | 1573788.1 (43.8%) | 1406924.7 (40.3%) | 1802775.0 (47.9%) | 2253367.8 (58.7%) | |
| 5–10 drinks/month | 765387.6 (21.3%) | 823183.4 (23.6%) | 773406.0 (20.6%) | 490251.1 (12.8%) | |
| 10+ drinks/month | 152334.3 (4.2%) | 215418.1 (6.2%) | 196026.9 (5.2%) | 180329.3 (4.7%) | |
| **Smoking status** | | | | | 0.667 |
| Never | 3189607.6 (88.8%) | 3119617.6 (89.3%) | 3430094.7 (91.2%) | 3466919.1 (90.4%) | |
| Current smoker | 394164.7 (11.0%) | 365557.0 (10.5%) | 321622.1 (8.6%) | 351387.7 (9.2%) | |
| Former smoker | 7807.1 (0.2%) | 9339.0 (0.3%) | 8151.4 (0.2%) | 18648.3 (0.5%) | |
| **Total calories (cal)** | 1695.49 (1355.80, 2140.65) | 1764.06 (1410.50, 2227.71) | 1823.38 (1403.04, 2262.81) | 1894.45 (1460.45, 2321.33) | 0.055 |
| **Recreational physical activity** | 2093124.8 (58.3%) | 1831674.4 (52.4%) | 1484616.5 (39.5%) | 1380887.9 (36.0%) | < 0.001 |
| **Diabetes** | 339178.7 (9.4%) | 475228.8 (13.6%) | 777117.8 (20.7%) | 1252314.3 (32.6%) | < 0.001 |
| **Stroke** | 214433.5 (6.0%) | 231651.6 (6.6%) | 232204.8 (6.2%) | 294146.0 (7.7%) | 0.740 |
| **Comorbidity index** | | | | | < 0.001 |
| 1 | 1339656.2 (37.3%) | 1374742.6 (39.3%) | 1592486.3 (42.4%) | 1726806.9 (45.0%) | |
| 2 | 1468050.1 (40.9%) | 1117512.2 (32.0%) | 982940.8 (26.1%) | 669895.2 (17.5%) | |
| 3 or greater | 783873.0 (21.8%) | 1002258.8 (28.7%) | 1184441.1 (31.5%) | 1440253.0 (37.5%) | |
| **Sleep disorder** | 220942.9 (6.2%) | 288324.0 (8.3%) | 452613.4 (12.0%) | 815257.5 (21.2%) | < 0.001 |

[†]Percentage and median (Q25, Q75) were weighted.

[*]P<0.05

**Table 3. Weighted logistic regression model of the effect of different levels of waist circumstance on sleep disorder.**

| Groups | SE | Wald | OR (95%CI) | P | P for trend |
|---|---|---|---|---|---|
| Model 1[a] | | | | | < 0.001 |
| Q1 | | | 1.00 | | |
| Q2 | 0.367 | 0.743 | 1.37 (0.65–2.91) | 0.396 | |
| Q3 | 0.351 | 4.404 | 2.09 (1.02–4.28) | 0.045 | |
| Q4 | 0.234 | 36.529 | 4.12 (2.55–6.64) | < 0.001 | |
| Model 2[b] | | | | | < 0.001 |
| Q1 | | | 1.00 | | |
| Q2 | 0.376 | 0.595 | 1.34 (0.62–2.90) | 0.447 | |
| Q3 | 0.360 | 3.615 | 1.98 (0.95–4.15) | 0.068 | |
| Q4 | 0.249 | 28.713 | 3.79 (2.28–6.33) | < 0.001 | |
| Model 3[c] | | | | | < 0.001 |
| Q1 | | | 1.00 | | |
| Q2 | 0.348 | 0.334 | 1.22 (0.58–2.58) | 0.572 | |
| Q3 | 0.331 | 2.276 | 1.65 (0.81–3.35) | 0.154 | |
| Q4 | 0.234 | 18.639 | 2.75 (1.66–4.54) | < 0.001 | |

[a]Unadjusted

[b]Adjusted for age, and gender

[c]Adjusted for age, gender, race, marital status, alcohol intake, recreational physical activity, diabetes, and comorbidity index.

## Subgroup analyses

Subgroup analyses for the association between different waist circumstance levels and incidence of sleep disorder. The participants were divided into subgroups according to age, gender and race. The results showed that the association between different waist circumference levels and incidence of sleep disorder stably existed in the different subgroups (Fig 3, $P_{trend} < 0.05$).

## Discussion

Using NHANES data, the present study sought to investigate the association between waist circumference and sleep disorder in the elderly population. Our research revealed that elderly participants with a sleep disorder had a substantially greater waist circumference than those without a sleep disorder. The results of this study revealed association between waist circumference and sleep disorder in older adults, providing support for the hypothesis that central adiposity may be a risk factor for sleep disturbances [3, 20, 21].

Moreover, our study revealed that participants in the higher quartiles of waist circumference had substantially greater odds of suffering from a sleep disorder than those in the lowest quartile. This suggests that central obesity may have a cumulative influence on the risk of sleep disturbances in the elderly. Even after controlling for potential confounding factors such as age, gender, and lifestyle, the observed association between waist circumference and sleep disorder remained significant. The results also showed a linear dose-response relationship between waist circumference and sleep disorder. With the increase of waist circumference, the OR of sleep disorder increased. This strengthens the reliability of our findings and provides support for the independent function of waist circumference in predicting sleep disorders in the elderly. Age and gender-based subgroup analyses did not reveal any significant differences in the association between waist circumference and sleep disorder among the elderly. This indicates that the association holds true across all demographic subgroups of the elderly population.

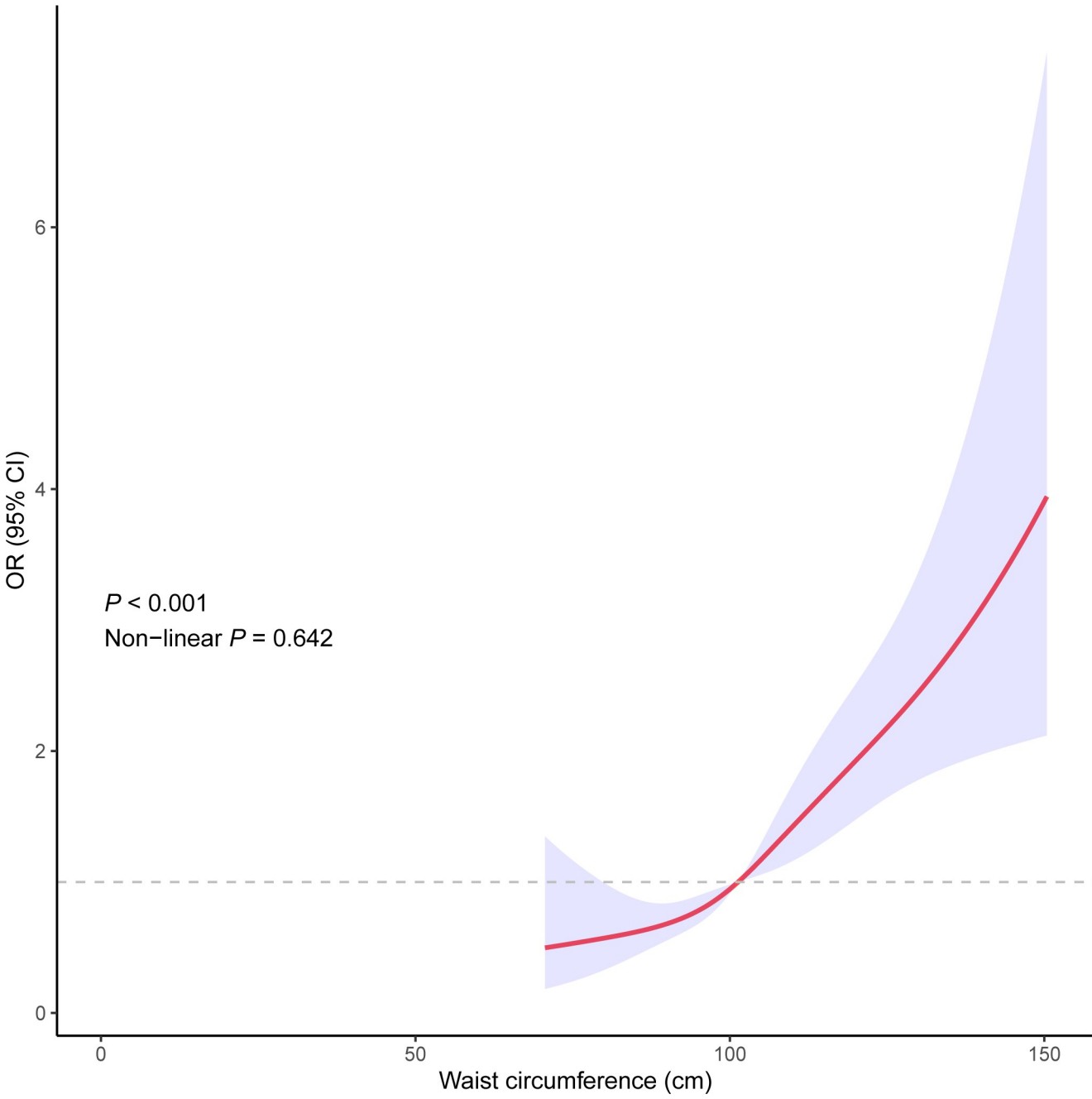

**Fig 2. Dose–response relationship between waist circumference and sleep disorder.**

Several plausible explanations can be proposed for the underlying mechanisms connecting waist circumference and sleep disorder, which are not completely understood. Previous studies have found that disrupted or disturbed sleep seems to contribute to the accumulation of body fat [22]. Visceral fat accumulation around the abdomen has been linked to an increased release of inflammatory cytokines and adipokines, which can disrupt sleep patterns and diminish sleep quality [23–26]. Central adiposity is also associated with metabolic abnormalities such as insulin resistance, dyslipidemia, and inflammation, which can disrupt the normal sleep-wake

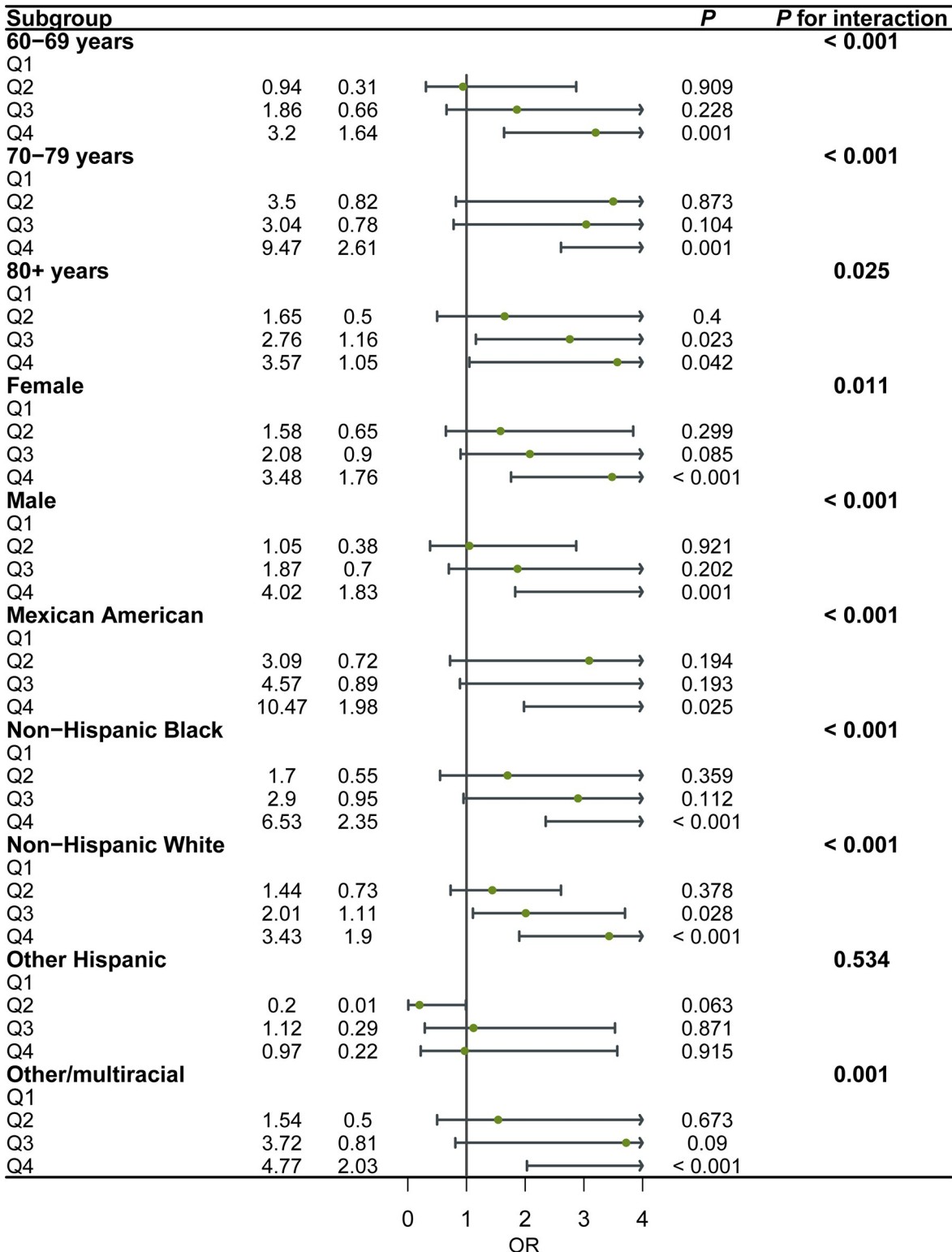

| Subgroup | | | | P | P for interaction |
|---|---|---|---|---|---|
| **60–69 years** | | | | | **< 0.001** |
| Q1 | | | | | |
| Q2 | 0.94 | 0.31 | | 0.909 | |
| Q3 | 1.86 | 0.66 | | 0.228 | |
| Q4 | 3.2 | 1.64 | | 0.001 | |
| **70–79 years** | | | | | **< 0.001** |
| Q1 | | | | | |
| Q2 | 3.5 | 0.82 | | 0.873 | |
| Q3 | 3.04 | 0.78 | | 0.104 | |
| Q4 | 9.47 | 2.61 | | 0.001 | |
| **80+ years** | | | | | **0.025** |
| Q1 | | | | | |
| Q2 | 1.65 | 0.5 | | 0.4 | |
| Q3 | 2.76 | 1.16 | | 0.023 | |
| Q4 | 3.57 | 1.05 | | 0.042 | |
| **Female** | | | | | **0.011** |
| Q1 | | | | | |
| Q2 | 1.58 | 0.65 | | 0.299 | |
| Q3 | 2.08 | 0.9 | | 0.085 | |
| Q4 | 3.48 | 1.76 | | < 0.001 | |
| **Male** | | | | | **< 0.001** |
| Q1 | | | | | |
| Q2 | 1.05 | 0.38 | | 0.921 | |
| Q3 | 1.87 | 0.7 | | 0.202 | |
| Q4 | 4.02 | 1.83 | | 0.001 | |
| **Mexican American** | | | | | **< 0.001** |
| Q1 | | | | | |
| Q2 | 3.09 | 0.72 | | 0.194 | |
| Q3 | 4.57 | 0.89 | | 0.193 | |
| Q4 | 10.47 | 1.98 | | 0.025 | |
| **Non–Hispanic Black** | | | | | **< 0.001** |
| Q1 | | | | | |
| Q2 | 1.7 | 0.55 | | 0.359 | |
| Q3 | 2.9 | 0.95 | | 0.112 | |
| Q4 | 6.53 | 2.35 | | < 0.001 | |
| **Non–Hispanic White** | | | | | **< 0.001** |
| Q1 | | | | | |
| Q2 | 1.44 | 0.73 | | 0.378 | |
| Q3 | 2.01 | 1.11 | | 0.028 | |
| Q4 | 3.43 | 1.9 | | < 0.001 | |
| **Other Hispanic** | | | | | **0.534** |
| Q1 | | | | | |
| Q2 | 0.2 | 0.01 | | 0.063 | |
| Q3 | 1.12 | 0.29 | | 0.871 | |
| Q4 | 0.97 | 0.22 | | 0.915 | |
| **Other/multiracial** | | | | | **0.001** |
| Q1 | | | | | |
| Q2 | 1.54 | 0.5 | | 0.673 | |
| Q3 | 3.72 | 0.81 | | 0.09 | |
| Q4 | 4.77 | 2.03 | | < 0.001 | |

OR

**Fig 3. Relationship between waist circumference and sleep disorder in subgroups of potential effect modifiers.**

cycle and result in sleep disturbances [27, 28]. In addition, excess abdominal fat can exert mechanical pressure on the diaphragm and airways, impairing respiratory function during sleep and contributing to conditions like sleep apnea [29, 30].

This study's findings have significant implications for the prevention and treatment of sleep disorders in the geriatric population. A straightforward and cost-effective method for identifying individuals at increased risk for sleep disturbances could be waist circumference measurement screening. Interventions that target abdominal obesity, such as lifestyle modifications and weight management programs, may enhance the quality of sleep and reduce the prevalence of sleep disorders in the elderly. Nonetheless, a number of limitations of this investigation must be acknowledged. Firstly, the study's cross-sectional design hinders our ability to establish a causal link between waist circumference and sleep disorder. To further examine the temporal relationship between these variables, longitudinal studies are required. Secondly, the evaluation of sleep disorders relied on self-reported questionnaires, which are susceptible to recall bias and subjectivity. Future research integrating objective measures of sleep quality, such as polysomnography, would yield more convincing results. As with any observational study, the presence of unmeasured confounding variables cannot be ruled out entirely. Thirdly, We cannot capture all confounding factors, so the scientific nature of the conclusions still needs to be verified by more rigorous experimental design.

## Conclusion

Our investigation demonstrates a significant correlation between waist circumference and sleep disorders in the elderly population. There was a dose-response relationship between waist circumference and the likelihood of suffering from a sleep disorder. These results highlight the significance of addressing central adiposity as a potential risk factor for sleep disturbances in the elderly. Further research is necessary to elucidate the underlying mechanisms and investigate effective interventions targeting abdominal adiposity to improve sleep quality and overall health in this vulnerable population.

## Author Contributions

**Conceptualization:** Kejun Tian.

**Data curation:** Yuting Zhong, Maolin Zhong, Cheng Peng, Hui Zhang, Kejun Tian.

**Formal analysis:** Yuting Zhong, Kejun Tian.

**Funding acquisition:** Ying Li.

**Investigation:** Yuting Zhong, Kejun Tian.

**Methodology:** Ying Li, Kejun Tian.

**Project administration:** Kejun Tian.

**Software:** Hui Zhang.

**Validation:** Ying Li, Maolin Zhong.

**Visualization:** Maolin Zhong, Cheng Peng.

**Writing – original draft:** Yuting Zhong.

**Writing – review & editing:** Kejun Tian.

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
