## [Decision Letter · Decision Letter 0]

3 Jun 2024

PONE-D-24-14233Association between Waist Circumference and Sleep Disorder in the Elderly: Based on the NHANES 2005–2018PLOS ONE

Dear Dr. Tian,

Thank you for submitting your manuscript to PLOS ONE. After careful consideration, we feel that it has merit but does not fully meet PLOS ONE’s publication criteria as it currently stands. Therefore, we invite you to submit a revised version of the manuscript that addresses the points raised during the review process.

We look forward to receiving your revised manuscript.

Kind regards,

Patricia Khashayar

Academic Editor

PLOS ONE

Journal Requirements:

**Additional Editor Comments:**

This is an interesting article but the reviewers have raised certain concerns. The authors should address these concerns before the article could be published

Reviewers' comments:

Reviewer's Responses to Questions

**Comments to the Author**

1. Is the manuscript technically sound, and do the data support the conclusions?

Reviewer #1: Partly

Reviewer #2: Yes

2. Has the statistical analysis been performed appropriately and rigorously? 

Reviewer #1: N/A

Reviewer #2: No

3. Have the authors made all data underlying the findings in their manuscript fully available?

Reviewer #1: Yes

Reviewer #2: Yes

4. Is the manuscript presented in an intelligible fashion and written in standard English?

Reviewer #1: No

Reviewer #2: Yes

5. Review Comments to the Author

Reviewer #1: 1. The dose-response relationship was mentioned in the summary conclusion, but I did not see the corresponding result explanation in the results, please add.

2. Pay attention to check whether the format of the paper meets the requirements of the journal.

3. It is suggested to adjust the expression of the paper so that the context is smooth and there is no language disorder. There are both "()" and "[]" in the table, which must be unified.

4.NHANES data is required to be weighted in the analysis, but I did not see any words related to weighting in the whole text, please add clarification. The results of the full text are unreliable if they are not weighted during the analysis.

5. The introduction part covers the severity of sleep disorders and the function of waist circumference indicators, indicating that waist circumference is related to various adverse health outcomes, but it cannot lead to the scientific hypothesis for the study of the correlation between waist circumference and sleep disorders in this paper. It is suggested to supplement the possible correlation between the two.

6. Both hypertension and diabetes exist as independent covariates in previous papers. Why did the author choose diabetes and stroke as independent variables and put hypertension together with other comorbidities? Is this reasonable? Please explain.

7. Many studies have shown that there may be a correlation between sleep disorders and depression, suggesting that the author describe depression as a covariable.

8. Existing studies have shown that serum cotinine is a marker of tobacco exposure, and it is recommended to replace the smoking covariate with serum cotinine.

9. Are sleep disorders related to sleep duration? There is no description of sleep duration in the selection of variables, please add clarification.

10. Suggestions in the table, P< 0.05 to be * and explain.

11. It is recommended to convert Table 4 into a forest map format, as the table looks too messy.

12.Model 3 notes without adjustments for stroke, education level, and Total calories? Is there any adjustment for BMI? The full text does not see the description of BMI, but the Model 3 notes involve BMI, is there a commonality between BMI and waist circumference?

Reviewer #2: Comments to the authors:

This study addresses an important health problem in the elderly. The study evaluates whether waist circumference (WC) is related to sleep disorder. The authors report a significant association between WC and sleep disorder particularly in higher quartiles of WC. The authors concluded that people above 60 years with larger WC were more likely to have a sleep disorder. Whilst this study is valuable, the below weaknesses need to be addressed to strengthen it.

Methods

Study population: Please provide a summary of the sample design NHANES uses for its data collection and what makes it nationally representative.

Lines 15 – 19: What was the difference in the characteristics of those with missing information and those without missing information? Address how the missing information of nearly 80% of the eligible participants above 60 years could influence your results? Will the results change if you imputed?

Line 21 – 22: Waist circumference couldn’t be a covariate but the main independent variable. It would be important to clarify that by having a separate heading for waist circumference and be clear it is the independent variable. Also provide information on the units of measurement and if any recategorization was done and why

Line 23 – 30; 1- 12: More information on how some covariates were measured but nothing was provided on covariates such as age, race, family income, recreational physical activity. Overall, this section needs some improvement including stating how the covariates were measured and categorized/recategorized.

Outcome:

How was the outcome treated? Yes or No? It seems there was information on the types of sleep disorders. Any plans for subgroup analyses with the different sleep disorders?

Lines: 11 – 12: Is the complication index a standardized index? Please provide reference if it was previously standardized. Otherwise, address the issues of standardization and validity for this index.

Statistical analysis:

Line 22: What was the basis for showing continuous variables as medians and IQRs?

Line 26: Why did you choose four categories for the WC? Any precedence for this approach?

Line 27 – 30: How did you select the confounders? How many variables were adjusted in the final model?

Line 1 – 2: How many age categories did you create for the subgroup analysis? How many racial groups did you use for subgroup analysis? And how many were collected? Any information on ethnicities?

The language in the analysis needs improvement.

Results:

Line 18: comorbidity index? Or complication index? You will need to provide more details in the methods section.

Lines 21 – 24: It is a better approach to present percentages than absolute numbers.

Association of WC with sleep disorder

Lines 17 – 21: There is lack of clarity on the sensitivity analysis. What were you testing for? Was there any suspected residual confounding by any unmeasured variable you used for the sensitivity analysis? P for trend is not a measure of sensitivity testing and not sure why that was mentioned here.

Subgroup analyses:

Lines 24 – 29: There was correlation analysis conducted here and so you may need to be mindful of the use of this term. Was there any evidence of effect modification by age, gender and race? You will need to present the results of the subgroup analyses other than the trend. Any evidence that the association differed by age, gender or race?

Discussion

Line 6 – 7: No correlation analysis was conducted, and the authors need to limit themselves to the correct terminology.

Lines 25 – 27: This is pointing to evidence of reverse causality. NHANES is a cross-sectional study and hence reverse causality is a big issue. How did you address this?

Table 3: No need to present both the betas and odds ratios information.

Table 4: Same comment as Table 4. Please present only the odds ratios. Also check the formatting of the Table

6. PLOS authors have the option to publish the peer review history of their article (what does this mean?). If published, this will include your full peer review and any attached files.

Reviewer #1: **Yes: **shan liu

Reviewer #2: No

---

## [Author Response · Author response to Decision Letter 0]

12 Jun 2024

RESPOND TO REVIEWERS: 

Reviewer 1

Comment 1: The dose-response relationship was mentioned in the summary conclusion, but I did not see the corresponding result explanation in the results, please add.

Reply 1: Thank you for your professional review. After consulting professional statisticians, we further evaluated the dose-response relationship by using restricted cubic spline plots, and the relevant results are shown in Figure 2. Dose-response relationship refers to that when the study factor can be quantified or graded, the change of the amount of the factor can affect the change of the morbidity of the population, and the possibility of causal relationship between the two is greater. That is, as the waist circumference of the study population increased, the incidence of sleep disorders also increased. (See page 8, line 25)

Comment 2. Pay attention to check whether the format of the paper meets the requirements of the journal.

Reply 2: Thank you for your professional review. We have revised the format of the manuscript to meet the requirements of the journal.

Comment 3. It is suggested to adjust the expression of the paper so that the context is smooth and there is no language disorder. There are both "()" and "[]" in the table, which must be unified.

Reply 3: Thanks for your careful review. We have revised this.

Comment 4.NHANES data is required to be weighted in the analysis, but I did not see any words related to weighting in the whole text, please add clarification. The results of the full text are unreliable if they are not weighted during the analysis.

Reply 4: Thank you for your professional review. We have consulted relevant literature in the previous research and found that similar literature has not carried out weighted analysis ([1] Hu PW, Yang BR, Zhang XL, Yan XT, Ma JJ, Qi C, Jiang GJ. The association between dietary inflammatory index with endometriosis: NHANES 2001-2006. PLoS One. 2023 Apr 26;18(4):e0283216. [2] Zheng D, Zhao C, Ma K, Ruan Z, Zhou H, Wu H, Lu F. Association between visceral adiposity index and risk of diabetes and prediabetes: Results from the NHANES (1999-2018). PLoS One. 2024 Apr 25;19(4):e0299285.). Therefore, we did not consider weighting in the analysis process, and future studies will definitely take this into account to make the conclusions of the paper more scientific. Hope to get your understanding.

Comment 5. The introduction part covers the severity of sleep disorders and the function of waist circumference indicators, indicating that waist circumference is related to various adverse health outcomes, but it cannot lead to the scientific hypothesis for the study of the correlation between waist circumference and sleep disorders in this paper. It is suggested to supplement the possible correlation between the two.

Reply 5: Thank you for your professional review. We have cited relevant literature in the Introduction part to prove the potential relationship between the two. Due to space problems, we focus on the relevant mechanism of the correlation between the two in the Discussion part. (See page 3, line 19)

Comment 6. Both hypertension and diabetes exist as independent covariates in previous papers. Why did the author choose diabetes and stroke as independent variables and put hypertension together with other comorbidities? Is this reasonable? Please explain.

Reply 6: Thank you for your professional review. What you said is very correct. When we consulted the relevant literature and analyzed, we accidentally found the comorbidities index, calculated the score and analyzed it. We consider that hypertension is very common and often co-exists with other comorbidities, such as cardiovascular disease, obesity, and metabolic syndrome. These conditions share common risk factors and may together contribute to the development of sleep disorders. Including high blood pressure as a separate variable may introduce redundancy, as it often accompanies diabetes and stroke. By grouping it with other comorbidities, the analysis can avoid multicollinearity and better isolate the unique effects of diabetes and stroke. Hope to get your approval.

Comment 7. Many studies have shown that there may be a correlation between sleep disorders and depression, suggesting that the author describe depression as a covariable.

Reply 7: Thank you for your professional review. Depression has been considered as one of the covariables in our analysis. Then, in the process of data cleaning, we found that there were many missing data of this index, so this index was abandoned for analysis. This will be taken into account in future studies. Thank you for your valuable advice.

Comment 8. Existing studies have shown that serum cotinine is a marker of tobacco exposure, and it is recommended to replace the smoking covariate with serum cotinine.

Reply 8: Thank you for your professional review. We acknowledge your comments. Firstly, Serum cotinine levels fluctuate based on recent nicotine exposure and have a relatively short half-life. This makes it a good marker for recent exposure but not for chronic exposure or long-term smoking behavior. Self-reported smoking status can give a more comprehensive picture of an individual's smoking history over months or years. Secondly, Collecting serum samples for cotinine analysis can be more invasive, costly, and logistically challenging than obtaining self-reported smoking information through questionnaires. This might not be feasible in all study settings, particularly in large-scale epidemiological studies or resource-limited settings. Finally, While serum cotinine is an objective measure and can help reduce misclassification due to underreporting or misreporting in self-reported data, it is not without limitations. Factors such as secondhand smoke exposure, nicotine replacement therapy, and individual metabolic differences can affect cotinine levels and potentially lead to misclassification of smoking status. However, in the future research, we will take your comments to improve the article. Hope you can understand.

Comment 9. Are sleep disorders related to sleep duration? There is no description of sleep duration in the selection of variables, please add clarification.

Reply 9: Thank you for your professional review. We have supplemented this in the manuscript. There are items on sleep duration in the questionnaire of related sleep disorders, but we refer to relevant literature and adopt the items that are more suitable for diagnosis for definition [Rahman HH, Niemann D, Yusuf KK. Association of urinary arsenic and sleep disorder in the US population: NHANES 2015-2016. Environ Sci Pollut Res Int. 2022 Jan;29(4):5496-5504. doi: 10.1007/s11356-021-16085-6. Epub 2021 Aug 22. PMID: 34420169.]. (See page 6, line 12)

Comment 10. Suggestions in the table, P< 0.05 to be * and explain.

Reply 10: Thanks for your good advice. We have revised it.

Comment 11. It is recommended to convert Table 4 into a forest map format, as the table looks too messy.

Reply 11: Thanks for your good advice. We have revised it.

Comment 12. Model 3 notes without adjustments for stroke, education level, and Total calories? Is there any adjustment for BMI? The full text does not see the description of BMI, but the Model 3 notes involve BMI, is there a commonality between BMI and waist circumference?Please consider using widely accepted diagnostic criteria for BMI to change the continuous variable into categorical variables. I believe that during clinical practice, a clear conclusion based on already known criteria would be easier to access instead of quartile numbers, such as underweight, normal weight, overweight, and obesity.

Reply 12: Thanks for your careful review. This is a mistake we made in writing the paper. BMI was not included in the analysis in this study, because waist circumference was found to be collinearity with BMI in the previous analysis, and waist circumference was found to be a better indicator of central obesity in other studies, and it was simple and easy to measure daily, so waist circumference was selected as a variable. 

Reviewer 2

This study addresses an important health problem in the elderly. The study evaluates whether waist circumference (WC) is related to sleep disorder. The authors report a significant association between WC and sleep disorder particularly in higher quartiles of WC. The authors concluded that people above 60 years with larger WC were more likely to have a sleep disorder. Whilst this study is valuable, the below weaknesses need to be addressed to strengthen it.

Methods

Comment 1. Study population: Please provide a summary of the sample design NHANES uses for its data collection and what makes it nationally representative.

Reply 1: Thank you for your professional review. We have supplemented the relevant information in the manuscript. (See page 4, line 14)

Comment 2. Lines 15 – 19: What was the difference in the characteristics of those with missing information and those without missing information? Address how the missing information of nearly 80% of the eligible participants above 60 years could influence your results? Will the results change if you imputed?

Reply 2: Thank you for your professional review. Indeed, in the process of data cleaning, we found a large number of data missing samples. We have consulted the relevant literature, and the basic processing method of the literature is to delete the samples of missing data. At that time, we also wanted to compare the feature data of the missing sample with that of the non-missing sample. However, due to the large number of missing data items, the scientific nature of the statistical results would be questioned. Therefore, we do not compare them. It must be that so many missing items have a great impact on the representativeness of the samples. However, as for the conclusions drawn from the included samples, it is hoped that more scientific methods will be adopted in the future to further promote the universality of the conclusions. Similar studies can be seen: Chen W, Sun X, Han J, Wu X, Wang Q, Li M, Lei X, Wu Y, Li Z, Luo G, Wei M. Joint effect of abnormal systemic immune-inflammation index (SII) levels and diabetes on cognitive function and survival rate: A population-based study from the NHANES 2011-2014. PLoS One. 2024 May 6;19(5):e0301300. doi: 10.1371/journal.pone.0301300. PMID: 38709763; PMCID: PMC11073711. Hu H, Wu Y, Zhao M, Liu J, Xie P. Sleep duration time and human papillomavirus infection risk: The U-shaped relationship revealed by NHANES data. PLoS One. 2024 Apr 5;19(4):e0301212. doi: 10.1371/journal.pone.0301212. PMID: 38578744; PMCID: PMC10997073. Hope to get your understanding.

Comment 3. Line 21 – 22: Waist circumference couldn’t be a covariate but the main independent variable. It would be important to clarify that by having a separate heading for waist circumference and be clear it is the independent variable. Also provide information on the units of measurement and if any recategorization was done and why

Reply 3: Thank you for your professional review. Learned a lot from your comments. We have revised it in the manuscript. (See page 5, line 1)

Comment 4. Line 23 – 30; 1- 12: More information on how some covariates were measured but nothing was provided on covariates such as age, race, family income, recreational physical activity. Overall, this section needs some improvement including stating how the covariates were measured and categorized/recategorized.

Reply 4: Thank you for your professional review. Learned a lot from your comments. We have revised it in the manuscript. (See page 5, line 8)

Outcome:

Comment 5. How was the outcome treated? Yes or No? It seems there was information on the types of sleep disorders. Any plans for subgroup analyses with the different sleep disorders?

Reply 5: Thank you for your professional question. This study referred to the relevant literature to set sleep disorders into two categories. Because we cannot accurately and effectively identify subgroups of sleep disorders from the raw data, it is not possible to perform a precise subgroup analysis ([1] Hu PW, Yang BR, Zhang XL, Yan XT, Ma JJ, Qi C, Jiang GJ. The association between dietary inflammatory index with endometriosis: NHANES 2001-2006. PLoS One. 2023 Apr 26;18(4):e0283216. [2] Zheng D, Zhao C, Ma K, Ruan Z, Zhou H, Wu H, Lu F. Association between visceral adiposity index and risk of diabetes and prediabetes: Results from the NHANES (1999-2018). PLoS One. 2024 Apr 25;19(4):e0299285.). However, your review has guided our approach, and we will work to improve the process in the future to ensure that the conclusions are scientific.

Comment 6. Lines: 11 – 12: Is the complication index a standardized index? Please provide reference if it was previously standardized. Otherwise, address the issues of standardization and validity for this index.

Reply 6: Thank you for your careful review. We obtained this indicator through relevant papers and applied it in this study, and relevant references have been supplemented in the manuscript. (See page 6, line 5)

Statistical analysis:

Comment 7. Line 22: What was the basis for showing continuous variables as medians and IQRs?

Reply 7: Thank you for your professional question. We refer to the statistics of similar studies and do some normal distribution tests. The continuity variables in the baseline indicators are basically skewed data, so they are presented in the form of median and quartile.

Comment 8. Line 26: Why did you choose four categories for the WC? Any precedence for this approach?

Reply 8: Thank you for your professional question. At present, we have not found a more suitable grouping standard for waist circumference indicators, so we refer to the statistical method of the corresponding article, divided the independent variables into four groups according to the quartile, and stratified waist circumference as far as possible to highlight the influence of waist circumference on the results.[Wang W, Lu X, Li Q, Chen D, Zeng W. The Relationship between Blood Lead Level and Chronic Pain in US Adults: A Nationwide Cross-Sectional Study. Pain Ther. 2023 Oct;12(5):1195-1208. doi: 10.1007/s40122-023-00535-9. Epub 2023 Jun 30. PMID: 37391620; PMCID: PMC10444925.]

Comment 9. Line 27 – 30: How did you select the confounders? How many variables were adjusted in the final model?

Reply 9: Thank you for your good question. Confounding factors were identified by reviewing the literature through a previous survey. The final model correction index was determined by the single-factor analysis results in Table 2, and the correction index was indicated in the annotations in Table 3.

Comment 10. Line 1 – 2: How many age categories did you create for the subgroup analysis? How many racial groups did you use for subgroup analysis? And how many were collected? Any information on ethnicities?

Reply 10: Thank you for good question. Nhanes included the race of the participants. We divided the age into three categories and the race into five categories for analysis.

Comment 11. The language in the analysis needs improvement.

Reply 11: Thank you for your careful review. We have revised it. (See page 6, line 24)

Results:

Comment 12. Line 18: comorbidity index? Or complication index? You will need to provide more details in the methods section.

Reply 12: Thank you for your careful review. It is comorbidity index. We have revised it. (See page 6, line 8)

Comment 13. Lines 21 – 24: It is a better approach to present percentages than absolute numbers.

Reply 13: Thank you for your good comment. We have not used percentages because the proportion of people in the groups is not very different because the classification is by quartile. Hope to get your understanding.

Association of WC with sleep disorder

Comment 14. Lines 17 – 21: There is lack of clarity on the sensitivity analysis. What were you testing for? Was there any suspected residual confounding by any unmeasured variable you used for the sensitivity analysis? P for trend is not a measure of sensitivity testing and not sure why that was mentioned here.

Reply 14: Thank you for your professional review. This is not our consideration, we have removed the

---

## [Decision Letter · Decision Letter 1]

11 Jul 2024

PONE-D-24-14233R1Association between Waist Circumference and Sleep Disorder in the Elderly: Based on the NHANES 2005–2018PLOS ONE

Dear Dr. Tian,

Thank you for submitting your manuscript to PLOS ONE. After careful consideration, we feel that it has merit but does not fully meet PLOS ONE’s publication criteria as it currently stands. Therefore, we invite you to submit a revised version of the manuscript that addresses the points raised during the review process.

We look forward to receiving your revised manuscript.

Kind regards,

Patricia Khashayar

Academic Editor

PLOS ONE

Additional Editor Comments:

Many of the critical comments pointed out by reviewer 1 is either not addressed or the justification is not acceptable. I understand that there are some limitations but these limitations should be addressed in a scientific way or acknowledged in the limitation section. It is possible that some studies have used unweighted NHANES data but this is not a justification, you should clarify why weighing the data was not needed for your data.

Reviewers' comments:

Reviewer's Responses to Questions

**Comments to the Author**

1. If the authors have adequately addressed your comments raised in a previous round of review and you feel that this manuscript is now acceptable for publication, you may indicate that here to bypass the “Comments to the Author” section, enter your conflict of interest statement in the “Confidential to Editor” section, and submit your "Accept" recommendation.

Reviewer #1: (No Response)

Reviewer #3: All comments have been addressed

2. Is the manuscript technically sound, and do the data support the conclusions?

Reviewer #1: No

Reviewer #3: Yes

3. Has the statistical analysis been performed appropriately and rigorously? 

Reviewer #1: No

Reviewer #3: Yes

4. Have the authors made all data underlying the findings in their manuscript fully available?

Reviewer #1: No

Reviewer #3: Yes

5. Is the manuscript presented in an intelligible fashion and written in standard English?

Reviewer #1: Yes

Reviewer #3: Yes

6. Review Comments to the Author

Reviewer #1: 1. For the issue of paper weighting, this is the requirement for the use of data on the NHANES website, and there are clear guidelines on how to weight the data.

2. Other questions I don't think the author did a good job of adjusting and revising my questions.

Reviewer #3: (No Response)

7. PLOS authors have the option to publish the peer review history of their article (what does this mean?). If published, this will include your full peer review and any attached files.

Reviewer #1: No

Reviewer #3: No

---

## [Author Response · Author response to Decision Letter 1]

22 Jul 2024

RESPOND TO REVIEWERS: 

Reviewer 1

Comment 1: For the issue of paper weighting, this is the requirement for the use of data on the NHANES website, and there are clear guidelines on how to weight the data.

Reply 1: Thank you for your professional review. We have appropriately weighted the NHANES data in our analysis in accordance with the guidelines provided on the NHANES website. Specific details on the weighting methodology used are included in the manuscript to ensure clarity and transparency. We have applied the recommended weights (e.g., WTDRD1/7) to account for the complex sampling design, which includes stratified, cluster, and multistage sampling with unequal probabilities. This approach ensures that our statistical analyses, including standard error and confidence interval estimates, are accurate and reliable.

Comment 2: Other questions I don't think the author did a good job of adjusting and revising my questions.

Reply 2: Thank you for your professional review. We appreciate your thorough review and the insights you have provided. We have done our best to make changes in accordance with your valuable comments, however, there are some limitations that cannot be well addressed. We hope to design more scientific experimental schemes in the future to further improve the scientificity of the research. Please kindly understand.

Additional Editor

Additional Editor Comments:

Many of the critical comments pointed out by reviewer 1 is either not addressed or the justification is not acceptable. I understand that there are some limitations but these limitations should be addressed in a scientific way or acknowledged in the limitation section. It is possible that some studies have used unweighted NHANES data but this is not a justification, you should clarify why weighing the data was not needed for your data.

Reply: Thank you. We have weighted and analyzed our nhanes data, making extensive revisions in the manuscript.

---

## [Editor Report · Decision Letter 2]

1 Aug 2024

Association between Waist Circumference and Sleep Disorder in the Elderly: Based on the NHANES 2005–2018

PONE-D-24-14233R2

Dear Dr. Tian,

We’re pleased to inform you that your manuscript has been judged scientifically suitable for publication and will be formally accepted for publication once it meets all outstanding technical requirements.

Kind regards,

Patricia Khashayar

Academic Editor

PLOS ONE
---

## [Editor Report · Acceptance letter]

5 Aug 2024

PONE-D-24-14233R2 

PLOS ONE

Dear Dr. Tian, 

I'm pleased to inform you that your manuscript has been deemed suitable for publication in PLOS ONE. Congratulations! Your manuscript is now being handed over to our production team.

Kind regards, 

on behalf of

Dr. Patricia Khashayar 

Academic Editor

PLOS ONE